# Unsupervised Discovery of 3D Physical Objects from Video

**Yilun Du**
MIT

**Kevin Smith**
MIT

**Tomer Ulman**
Harvard University

**Joshua Tenenbaum**
MIT

**Jiajun Wu**
Stanford University

## Abstract

We study the problem of unsupervised physical object discovery. While existing frameworks aim to decompose scenes into 2D segments based off each object's appearance, we explore how physics, especially object interactions, facilitates disentangling of 3D geometry and position of objects from video, in an unsupervised manner. Drawing inspiration from developmental psychology, our Physical Object Discovery Network (POD-Net) uses both multi-scale pixel cues and physical motion cues to accurately segment observable and partially occluded objects of varying sizes, and infer properties of those objects. Our model reliably segments objects on both synthetic and real scenes. The discovered object properties can also be used to reason about physical events.

## 1 Introduction

From early in development, infants impose structure on their world. When they look at a scene, infants do not perceive simply an array of colors. Instead, they scan the scene and organize the world into objects that obey certain physical expectations, like traveling along smooth paths or not winking in and out of existence (Spelke & Kinzler, 2007; Spelke et al., 1992). Here we take two ideas from human, and particularly infant, perception for helping artificial agents learn about object properties: that coherent object motion constrains expectations about future object states, and that foveation patterns allow people to scan both small or far-away and large or close-up objects in the same scene.

Motion is particularly crucial in the early ability to segment a scene into individual objects. For instance, infants perceive two patches moving together as a single object, even though they look perceptually distinct to adults (Kellman & Spelke, 1983). This segmentation from motion even leads young children to expect that if a toy resting on a block is picked up, both the block and the toy will move up as if they are a single object. This suggests that artificial systems that learn to segment the world could be usefully constrained by the principle that there are objects that move in regular ways.

In addition, human vision exhibits foveation patterns, where only a local patch of a scene is often visible at once. This allows people to focus on objects that are otherwise small on the retina, but also stitch together different glimpses of larger objects into a coherent whole.

We propose the Physical Object Discovery Network (POD-Net), a self-supervised model that learns to extract object-based scene representations from videos using motion cues. POD-Net links a visual generative model with a dynamics model in which objects persist and move smoothly. The visual generative model factors an object-based scene decompositions across local patches, then aggregates those local patches into a global segmentation. The link between the visual model and the dynamics model constrains the discovered representations to be usable to predict future world states. POD-Net thus produces more stable image segmentations than other self-supervised segmentation models, especially in challenging conditions such as when objects occlude each other (Figure 1).

We test how well POD-Net performs image segmentation and object discovery on two datasets: one made from ShapeNet objects (Chang et al., 2015), and one from real-world images. We find that POD-Net outperforms recent self-supervised image segmentation models that use regular foreground-background relationships (Greff et al., 2019) or assume that images are composable into object-like parts (Burgess et al., 2019). Finally, we show that the representations learned by POD-Net can be used to support reasoning in a task that requires identifying scenes with physically implausible events

---

Project page: https://yilundu.github.io/podnet

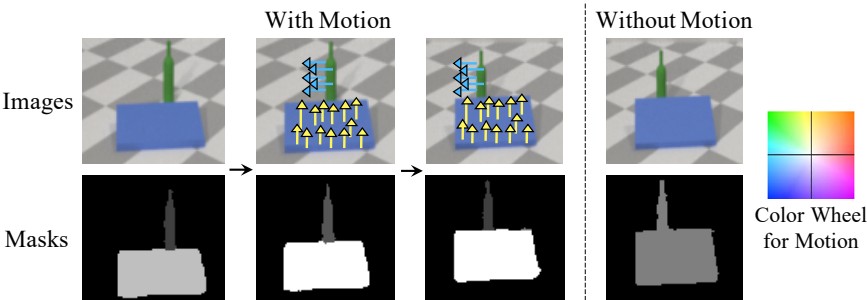

Figure 1: Motion is an important cue for object segmentation from early in development. We combine motion with an approximate understanding of physics to discover 3D objects that are physically consistent across time. In the video above, motion cues (shown with colored arrows) enable our model to modify our predictions from a single large incorrect segmentation mask to two smaller correct masks.

(Smith et al., 2019). Together, this demonstrates that using motion as a grouping cue to constrain the learning of object segmentations and representations achieves both goals: it produces better image segmentations and learns scene representations that are useful for physical reasoning.

## 2  RELATED WORK

Developing a factorized scene representation has been a core research topic in computer vision for decades. Most learning-based prior works are supervised, requiring annotated specifications such as segmentations (Janner et al., 2018), patches (Fragkiadaki et al., 2015), or simulation engines (Wu et al., 2017; Kansky et al., 2017). These supervised approaches face two challenges. First, in practical scenarios, annotations are often prohibitively challenging to obtain: we cannot annotate the 3D geometry, pose, and semantics of every object we encounter, especially for deformable objects such as trees. Second, supervised methods may not generalize well to out-of-distribution test data such as novel objects or scenes.

Recent research on unsupervised object discovery and segmentation in machine learning has attempted to address these issues: researchers have developed deep nets and inference algorithms that learn to ground visual entities with factorized generative models of static (Greff et al., 2017; Burgess et al., 2019; Greff et al., 2019; Eslami et al., 2016) and dynamic (van Steenkiste et al., 2018; Veerapaneni et al., 2019; Kosiorek et al., 2018; Eslami et al., 2018) scenes. Some approaches also learn to model the relations and interactions between objects (Veerapaneni et al., 2019; Stanić & Schmidhuber, 2019; van Steenkiste et al., 2018). The progress in the field is impressive, though these approaches are still mostly restricted to low-resolution images and perform less well on small or heavily occluded objects. Because of this, they often fail to observe key concepts such as object permanence and solidity. Furthermore, these models all segment objects in 2D, while our POD-Net aims to capture the 3D geometry of objects in the scene.

Some recent papers have integrated deep learning with differentiable rendering to reconstruct 3D shapes from visual data without supervision, although they mostly focused on images of a single object (Rezende et al., 2016; Sitzmann et al., 2019), or require multiview data as input (Yan et al., 2016). In contrast, we use object motion and physics to discover objects in 3D with physical occupancy. This allows our model to do better in both object discovery and future prediction, captures notions such as object permanence, and better aligns with people's perception, belief, and surprise signals of dynamic scenes. A separate body of work utilizes motion cues to segment objects (Brox & Malik, 2010; Bideau et al., 2018; Xie et al., 2019; Dave et al., 2019). Such works typically assume a single foreground object moving, and aggregate motion information across frames to segment out objects or separate moving parts of objects. Our work instead seeks to distill information captured from motion to discover objects in 3D from images.

Others works have explored 3D object discovery using RGB-D or 3D volumetric inputs (Herbst et al., 2011; Karpathy et al., 2013; Ma & Sibley, 2014). The presence of 3D information, such as depth, is a significant difference from our work. Such information allows approaches to reliably detect surface orientations and discontinuities (Karpathy et al., 2013; Herbst et al., 2011) which significantly reduces the difficulty of discovering objects, especially in the tabletop settings considered.

Our work is also related to research in computer vision on unsupervised object discovery from video (Lu et al., 2019; Wang et al., 2019; Yang et al., 2019b). Such works focus on detecting objects in

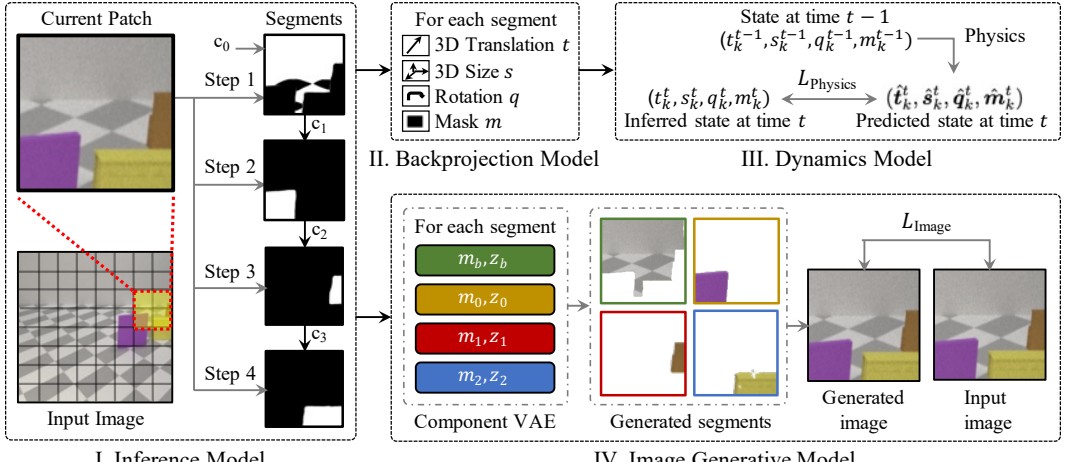

Figure 2: POD-Net contains four modules for discovering physical objects from video. (I) An inference model auto-regressively infers a set of candidate object masks and latents to describe each patch of an image; (II) A backprojection model maps each mask to a 3D primitive; (III) A dynamics model captures the motion of 3D physical objects; and (IV) An image generative model decodes latents and masks to reconstruct the image.

realistic videos, but only focus on detecting a single foreground object, instead of all component objects in a scene. Furthermore, such approaches rely on supervised information such as ImageNet weights, or pretrained segmentation networks for object detection, limiting their applicability to new objects in novel classes. Our our approach also assumes supervised information on the underlying 2D to 3D mapping, but it does not assume any supervision for object detection. We show that this enables our approach to generalize to novel ShapeNet objects.

## 3 METHOD

The Physical Object Discovery Network (POD-Net) (Figure 2) decomposes a dynamic scene into a set of component 3D physical primitives. POD-Net contains an inference model, which recursively infers a set of component primitive descriptions, masks, and latent vectors (Section 3.1). It also contains a three-module generative model (Section 3.2). The generative model consists of a back-projection module to infer 3D properties of each component, a dynamics model to predict primitives motions and a image generative model in the form a VAE (Kingma & Welling, 2014; Rezende et al., 2014) to render primitives onto 2D images. These components ensure that learned primitive representations can reconstruct the original image in a physically consistent manner. Together, these constraints produce a strong signal for self-supervised learning of object-centric scene representations.

### 3.1 INFERENCE MODEL

We sequentially infer the underlying masks and latents that represent a scene (Figure 2-I). Inspired by MONet (Burgess et al., 2019), we employ an attention network $\text{Attention}(\cdot)$ to decompose a scene into a set of separate masks $M = \{m_1, m_2, ..., m_n\}$. We extract a corresponding latent $z_i$ per mask $m_i$. In particular, we initialize context $c_0 = 1$, which we define to represent the context in the image $x$ yet to be explained. At each step, we decode the attention mask $m_i = c_{i-1}\text{Attention}(x; c_{i-1})$. We iteratively update the corresponding context in the image by $c_i = c_{i-1}(1 - \text{Attention}(x; c_{i-1}))$ to ensure that sum of all masks explain the entire image.

We further train a VAE encoder $\text{Encode}(z|m, x)$, which infers latents $z_i$ from each component mask $m_i$. We set $m_0, z_0$ – the first decoded mask and latent – to be the background mask $m_b$ and latent $z_b$, and define each subsequent mask or latent to be object masks and latents.

**Sub-patch decomposition.** Direct inference of component objects and background from a single image can be difficult, especially when images are complex and when objects are of vastly different sizes. An inference network must learn to pay attention to coarse features in order to segment large objects, and to fine details in the same image order to segment the smaller objects. Inspired by how people solve this problem by stitching together multiple foveations into a coherent whole, we train our models and apply inference on overlapping sub-patches of an image (Figure 3).

In particular, given an image of size $H \times W$, we divide the image into a $8 \times 8$ grid (pictured in the left of Figure 3), with each grid element having size $H/8 \times W/8$. We construct a sub-patch for every $2 \times 2$ component sub-grid, leading to a total of 64 different overlapping sub-patches. We apply inference on each sub-patch. Under this decomposition, smaller objects still

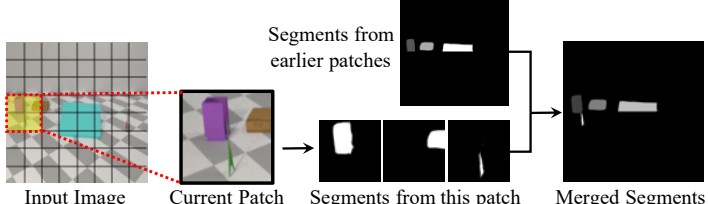

Segments from earlier patches

Input Image    Current Patch    Segments from this patch    Merged Segments

Figure 3: Illustration of sub-patch decomposition for image inference. An image is divided in a 8×8 grid, with inference is applied to each 2×2 sub-grid. To generate a global segmentation mask, object masks are sequentially inferred for each subpatch. Each object mask is either matched to an existing object or used to create a new object.

appear large in each sub-patch, while larger objects are shared across sub-patch.

To obtain a global segmentation map, we merge each sub-patch sequentially using a sliding window (Figure 3). At each step, we iterate through each segment given by the inference model from a sub-patch, and merge it with segments obtained from previous sub-patches, if there is an overlap in masks above 20 pixels. Every segment that does not get merged is initialized as a new object.

## 3.2 GENERATIVE MODEL

Our generative model represents a dynamic scene as a set of $K$ different physical objects and the surrounding background at each time step $t$. Each physical object $k$ is represented by its back-projection on 2D, a segmentation mask $\boldsymbol{m}_k^t \in \mathbb{R}^{HxW}$ of height $H$ and width $W$, and a latent code $\boldsymbol{z}_k^t \in \mathbb{R}^D$ of dimension $D$ for its appearance. In addition, the background is captured as a surrounding segmentation mask $\boldsymbol{m}_b^t \in \mathbb{R}^{HxW}$ and code $\boldsymbol{z}_b^t \in \mathbb{R}^D$. Segmentation masks are defined such that the sum of all masks corresponds to the entire image $\sum_k \boldsymbol{m}_k^t + \boldsymbol{m}_b^t = 1$.

We use a backprojection model to map segmentation masks $\boldsymbol{m}_k^t$ to 3D primitive cuboids (Figure 2-II). Cuboids are a coarse geometric representation that enable physical simulation. We next construct a dynamics model over the physical movement of predicted primitives (Figure 2-III). We further construct a generative model over images $\boldsymbol{x}^t$ by decoding latents $\boldsymbol{z}^t$ component-wise (Figure 2-IV).

**Backprojection model.** Our backprojection model maps a mask $\boldsymbol{m}_k$ to an underlying 3D primitive cuboid, represented as a translation $\boldsymbol{t}_k \in \mathbb{R}^3$, size $\boldsymbol{s}_k \in \mathbb{R}^3$, and rotation $\boldsymbol{q}_k \in \mathbb{R}^3$ (as a Euler angle) transform on a unit cuboid in a fully differentiable manner. In our case, we primarily pre-train a neural network to approximate the 2D-to-3D projection and use it as our differentiable backprojection model. However we show that such a task can also be approximated by assuming the camera parameters and the height of the plane is given (ignoring size and rotation regression), with little reduction in performance (see Appendix A.2 for further details).

**Dynamics model.** We construct a dynamics model over the next state of different physical objects $(\boldsymbol{t}_k^t, \boldsymbol{s}_k^t, \boldsymbol{q}_k^t, \boldsymbol{m}_k^t)$ by using first order approximation of velocity/angular velocity of the states of the object. Specifically, our model predicts

$$\hat{\boldsymbol{t}}_k^t = \boldsymbol{t}_k^{t-1} + \frac{1}{t-1}\sum_{i=1}^{t-1}(\boldsymbol{t}_k^i - \boldsymbol{t}_k^{i-1}), \quad \hat{\boldsymbol{s}}_k^t = \frac{1}{t}\sum_{i=0}^{t-1}\boldsymbol{s}_k^i, \tag{1}$$

$$\hat{\boldsymbol{q}}_k^t = \boldsymbol{q}_k^{t-1} + \frac{1}{t-1}\sum_{i=1}^{t-1}(\boldsymbol{q}_k^i - \boldsymbol{q}_k^{i-1}), \quad \hat{\boldsymbol{m}}_k^t = \text{Render}(\hat{\boldsymbol{t}}_k^t, \hat{\boldsymbol{s}}_k^t, \hat{\boldsymbol{q}}_k^t). \tag{2}$$

Our Render($\cdot$) function computes the segmentation mask of a predicted physical object, assuming all other physical objects are rendered. To compute this, we initialize a palette $\boldsymbol{p}_0 = \boldsymbol{1}$, which we define to represent the context in an image that has not been rendered yet. We further utilize a separate pre-trained model Project($\cdot$) that projects each primitive in 3D to a 2D segmentation mask (the inverse of the backprojection model described above). We then reorder predicted physical objects in increasing distance from the camera to $(\hat{\boldsymbol{t}}_{k'}^t, \hat{\boldsymbol{s}}_{k'}^t, \hat{\boldsymbol{q}}_{k'}^t)$. We then sequentially render each predicted physical object using Render($\hat{\boldsymbol{t}}_{k'}^t, \hat{\boldsymbol{s}}_{k'}^t, \hat{\boldsymbol{q}}_{k'}^t) = \boldsymbol{p}_{k'-1}\text{Project}(\hat{\boldsymbol{t}}_{k'}^t, \hat{\boldsymbol{s}}_{k'}^t, \hat{\boldsymbol{q}}_{k'}^t)$, and update the corresponding palette to be rendered as $\boldsymbol{p}_{k'} = \boldsymbol{p}_{k'-1}(1 - \text{Project}(\hat{\boldsymbol{t}}_{k'}^t, \hat{\boldsymbol{s}}_{k'}^t, \hat{\boldsymbol{q}}_{k'}^t))$.

Given modeled future states, the overall likelihood of a physical object $(\boldsymbol{t}_k^t, \boldsymbol{s}_k^t, \boldsymbol{q}_k^t, \boldsymbol{m}_k^t)$ is given by

$$p(\boldsymbol{t}_k^t, \boldsymbol{s}_k^t, \boldsymbol{q}_k^t, \boldsymbol{m}_k^t) = \mathcal{N}(\boldsymbol{t}_k^t; \hat{\boldsymbol{t}}_k^t, \sigma_t^2)\mathcal{N}(\boldsymbol{s}_k^t; \hat{\boldsymbol{s}}_k^t, \sigma_s^2)\mathcal{N}(\boldsymbol{q}_k^t; \hat{\boldsymbol{q}}_k^t, \sigma_q^2)p(\boldsymbol{m}_k^t, \hat{\boldsymbol{m}}_k^t), \tag{3}$$

where we assume Gaussian distributions over translation, sizes, and rotations with $\sigma_s = \sigma_t = \sigma_q = 1$. $p(\cdot)$ is the probability of a predicted mask, defined as $p(\hat{\boldsymbol{m}}_k^t, \boldsymbol{m}_k^t) = \mathbb{1}_{\boldsymbol{m}_k^t > 0.5}\hat{\boldsymbol{m}}_k^t + (1 - \mathbb{1}_{\boldsymbol{m}_k^t > 0.5})(1 - \hat{\boldsymbol{m}}_k^t)$, where $\mathbb{1}_{(\cdot)}$ is the indicator function on each individual pixel. Overall, our dynamics model seeks to enforce that objects have zero order motion and maintain shape.

**Image generative model.** We represent images $\boldsymbol{x}^t \in \mathbb{R}^D$ at each time step as spatial Gaussian mixture models, with each mixture model being defined by a segmentation mask $m$ in $\boldsymbol{M}$ (Section 3.1). Each corresponding latent $\boldsymbol{z}_k$ is decoded to a pixel-wise mean $\mu_{ik}$ and a pixel-wise mask prediction $d_{ik}$ using a VAE decoder $\text{Decode}(\mu_k, \boldsymbol{d}_k | \boldsymbol{z}_k)$. We assume each pixel $i$ is independent conditioned on $\boldsymbol{z}$, so that the likelihood becomes

$$p(\boldsymbol{x}|\boldsymbol{z}) = \prod_{i=0}^{D}\left(\sum_{k=1}^{K}\left(m_{ik}\mathcal{N}(x_i; \mu_{ik}, \sigma^2) \times p_\theta(d_{ik}|\boldsymbol{z}_k)\right) + m_{ib}\mathcal{N}(x_i; \mu_{ib}, \sigma_b^2) \times p_\theta(d_{ib}|\boldsymbol{z}_b)\right) \tag{4}$$

for background component $m_b$, $\mu_b$, $d_b$ and object components $m_k$, $\mu_k$, $d_k$, where $p_\theta(d_{ik}|\boldsymbol{z}_k) = p_\theta(d_{ik} = m_{ik}|\boldsymbol{z}_k)$, is the probability that decoded mask from the latent matches the ground truth mask for the mixture. We use $\sigma = 0.11$ and $\sigma_b = 0.07$ to break symmetry between object and background components, encouraging the background to model the more uniform image components (Burgess et al., 2019). Our overall loss encourages the decomposition of an image into a set of reusable sub-components, as well as a large background.

## 3.3 Training Loss

Our overall system is trained to maximize the likelihood of both physical object and image generative models. Our loss consists of $\mathcal{L}(\boldsymbol{x}^t) = \mathcal{L}_{\text{Physics}} + \mathcal{L}_{\text{Image}} + \mathcal{L}_{\text{KL}}$, maximizing the likelihood of physical dynamics, images, and variational bound. Our image loss is defined to be $\mathcal{L}_{\text{Image}} = -\log(p(\boldsymbol{x}^t|\boldsymbol{z}))$, based on Eqn. 4. Our physics loss is defined to be $\mathcal{L}_{\text{Physics}} = -\sum_{k=1}^{K}\log p(\boldsymbol{t}_k^t, \boldsymbol{s}_k^t, \boldsymbol{q}_k^t, \boldsymbol{m}_k^t)$, based on Eqn. 3, which enforces that decoded primitives are physically consistent. The KL loss is

$$\begin{aligned}\mathcal{L}_{\text{KL}} = &\beta\left(\sum_{k=1}^{K}\text{KL}(\text{Encode}(\boldsymbol{z}_k^t|\boldsymbol{x}^t, \boldsymbol{m}_k^t) \mid\mid p(z)) + \text{KL}(\text{Encode}(\boldsymbol{z}_b^t|\boldsymbol{x}^t, \boldsymbol{m}_b^t) \mid\mid p(z))\right) + \\ &\gamma\left(\sum_{k=1}^{K}\text{KL}(q_\psi(\boldsymbol{d}_k|\boldsymbol{x}) \mid\mid p_\theta(\boldsymbol{d}_k|\boldsymbol{z}_k)) + \text{KL}(q_\psi(\boldsymbol{d}_b|\boldsymbol{x}) \mid\mid p_\theta(\boldsymbol{d}_b|\boldsymbol{z}_b))\right),\end{aligned} \tag{5}$$

enforcing the variational lower bound on likelihood (Kingma et al., 2014), where for brevity, we use $q_\psi(\boldsymbol{d}_k|\boldsymbol{x})$ to represent the mask generation process in Section 3.1, and $p(z) = \mathcal{N}(0, 1)$ is the prior.

Our training paradigm consists of two different steps. We first maximize the likelihood of the model under the image generation objective. After qualitatively observing object-like masks (roughly after 100,000 iterations), we switch to maximizing the likelihood of the model under both the generation and physical plausibility objectives. Alternatively, we found that switching at loss convergence also worked well. We find that enforcing physical consistency during early stages of training detrimental, as the model has not discovered object-like primitives yet. We use the RMSprop optimizer with a learning rate of $10^{-4}$ within the PyTorch framework (Paszke et al., 2019) to train our models.

## 4 Evaluation

We evaluate POD-Net on unsupervised object discovery in two different scenarios: a synthetic dataset consisting of various moving ShapeNet objects, and a real dataset of block towers falling. We also test how inferred 3D primitives can support more advanced physical reasoning.

### 4.1 Moving ShapeNet

We use ShapeNet objects to explore the ability of POD-Net to learn to segment objects from appearance and motion cues. We also test its ability to generalize to new shapes and textures.

**Data.** To train models on moving ShapeNet objects, we use the generation code provided in the ADEPT dataset in Smith et al. (2019). We generate a training set of 1,000 videos, each 100 frames long, of objects (80% of the objects from 44 ShapeNet categories) as well as rectangular occluders. Objects move in either a straight line, back and forth, or rotate, but do not collide with each other.

**Setup.** Videos have a resolution of 1024×1024 pixels. We apply our model with a patch size of 256×256. We use a residual architecture (He et al., 2015) for the attention and VAE components.

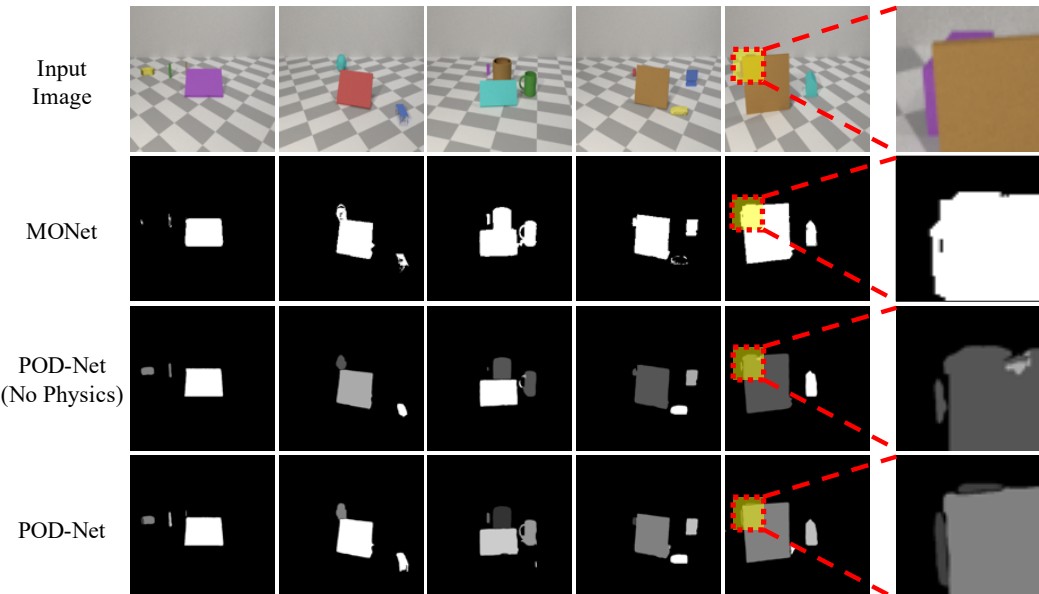

Figure 4: Comparisons of unsupervised object segmentation of POD-Net with and without motion and with MONet on scenes with synthetic objects. MONet is unable to seperate individual instances of objects, but is capable of getting a foreground mask of objects in a scene. POD-Net (no physics) is able to reliably detect almost all objects, though some instances of objects are merged together into a single object. POD-Net is able to reliably detect separate objects even when they are mostly occluded (zoomed-in images on right).

| Model | Multi-Scale | Phys | IoU | Detection | 3D IOU | 3D Recall |
|---|---|---|---|---|---|---|
| MONET | - | - | 0.289 (0.007) | 0.306 (0.005) | 0.019 (0.007) | 0.057 (0.028) |
| OP3 | - | - | 0.145 (0.004) | 0.121 (0.007) | 0.001 (0.001) | 0.000 (0.000) |
| Norm. Cuts | - | - | 0.634 (0.020) | 0.768 (0.029) | 0.034 (0.003) | 0.042 (0.005) |
| UVOD | - | - | 0.129 (0.006) | 0.062 (0.006) | 0.001 (0.000) | 0.000 (0.000) |
| Crisp Boundary Detection | - | - | 0.645 (0.012) | 0.727 (0.020) | 0.080 (0.004) | 0.020 (0.001) |
| POD-Net | No | No | 0.314 (0.010) | 0.361 (0.007) | 0.052 (0.012) | 0.171 (0.012) |
| POD-Net | No | Yes | 0.462 (0.007) | 0.512 (0.009) | 0.071 (0.012) | 0.287 (0.016) |
| POD-Net | Yes | No | 0.649 (0.011) | 0.709 (0.016) | 0.068 (0.011) | 0.251 (0.014) |
| POD-Net (Manual) | Yes | Yes | 0.685 (0.017) | 0.760 (0.016) | 0.090 (0.015) | 0.328 (0.016) |
| POD-Net | Yes | Yes | **0.739 (0.011)** | **0.821 (0.015)** | **0.095 (0.012)** | **0.374 (0.017)** |

Table 1: Average IoU of segmentations on the ADEPT dataset and the proportion of objects detected, where one segmentation mask has greater than 0.5 IoU, as well as average 3D IoU and recall. Standard error in parentheses.

Our backprojection model is pretrained on scenes of a single ShapeNet object, varied across different locations on a plane, with different rotations, translations, and scales. Our backprojection model only serves as a rough relative map from 2D mask to corresponding 3D position/size, as the dataset they are trained on utilize separate camera extrinsics/intrinsics than the ADEPT dataset, and also do not exhibit occlusions. To compute the physical plausibility $L_{physics}$ of primitives, we utilize the observations from the last three time steps. For efficiency, we evaluate physical plausibility on each component sub-patch of image. We train a recurrent model with a total of 5 slots for each image. Image segmentation is trained and evaluated on a per frame basis.

**Metrics.** To quantify our results, we measure the intersection over union (IoU) between the predicted segmentation masks and the corresponding ground truth masks. We compute the IoU for each ground truth mask by finding the maximum IoU intersection with a predicted mask. We report the average IoU across all objects in an image, as well as the percentage of objects detected in an image (with IoU > 0.5). To measure 3D inference ability, we report the maximum 3D IoU intersection between each ground truth 3D box and our inferred 3D bounding box. We also report the recall of ground truth 3D objects (Georgakis et al., 2016) detected in an image (with 3D IoU threshold > 0.1). We utilize our backprojection model to extract 3D bounding box proposals from 2D segmentations and apply a linear transformation to align coordinate spaces.

**Baselines.** We compare with two recent models of self-supervised object discovery, OP3 (Veerapaneni et al., 2019) and MONet (Burgess et al., 2019), as well as three algorithms for object

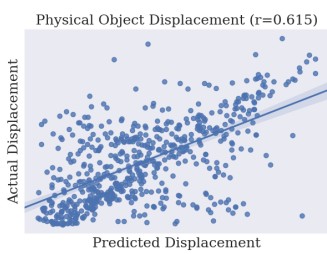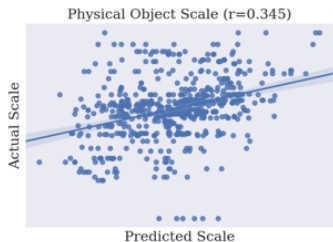

Figure 5: Visualization of discovered 3D primitives in two different scenes (left and right) through time. Our model is able to discover a 3D shape that is consistent with observed inputs under a perspective map. Furthermore, discovered primitive move coherently through time.

Figure 6: Plot of predicted translation of 3D primitive vs ground truth translation of 3D primitives (left) and plot of predicted scale of 3D primitives vs ground scale of 3D primitives (right).

segmentation, Normalized Cuts (Shi & Malik, 2000), Crisp Boundary (Isola et al., 2014), and the recent UVOD (Yang et al., 2019a). We train OP3 with 7 slots, with 4 steps of optimization per mask in the first image, and an additional step of optimization per future timestep. Due to memory constraints, we were only able to train the OP3 model on inputs of size 128 by 128. We train MONet on inputs of size 256 by 256. We apply normalized cuts on a region adjacency graph of the 256 by 256 image, and train UVOD in 256 by 256 images . We also compare with ablations of POD-Net: POD-Net applied directly to an image (single-scale) as opposed to across patches (multi-scale), POD-Net without physics, and POD-Net with a hard-coded backprojection model ('Manual') .

**Results.** We quantitatively compare object masks discovered by our model and other baselines in Table 1. We find that OP3 performs poorly, as it only discovers a limited subset of objects. MONet performs better and is able to discover a single foreground mask of all objects. However, the masks are not decomposed into separate component objects in a scene (Figure 4, 2nd row). Our scenes consist of a variable set of objects of vastly different scales, making it hard for MONet to learn to assign individual slots for each object. We find that a baseline based on normalizing cuts/crisp boundary detection is also able to segment objects, but is unable to get sharp segmentation boundaries for each object, and often decomposes a single object into multiple subobjects (see Appendix A.1 for details). Finally, UVOD also only segments a single foreground object.

We find that applying POD-Net (single scale, no physics) improves on MONet slightly, discovering several different masks containing multiple objects, albeit sometime missing other objects. POD-Net (single scale, physics) more reliably segments separate objects, but still misses objects. POD-Net (multi scale, no physics) reliably segments all objects in the scene, but often merges multiple objects into one object, especially when objects are overlapping (e.g., Figure 4, 3rd row). Finally, POD-Net obtains the best performance and segments all objects in the scene and individual objects where multiple objects overlap with each other (Figure 4, 4th row). Utilizing a manually coded backprojection module, POD-Net ('Manual') only leads to slight degradation in performance.

Next we analyze the 3D objects discovered by POD-Net. In Table 1, POD-Net performs the best, achieving the highest 3D average IoU intersection and recall. Crisp boundary detection obtains a high average IoU but low recall due to a large number of proposals. All IoUs are low due to the challenging nature of the task – obtaining high 3D IoU requires correct regression of size, position, and depth (using only RGB inputs). Even recent supervised 3D reconstruction approaches using 0.25 IoU thresholds for evaluation (Najibi et al., 2020). Visualizations of the discovered objects in Figure 5 show that POD-Net is able to segment a scene into a set of *temporally consistent* 3D cuboid primitives. We further find a high correlation $r = 0.615$ between the predicted and ground truth translation of objects and show plots of correlations in Figure 6 of ground truth and predicted objects.

**Generalization.** Just as young children can detect and reason about new objects with arbitrary shapes and colors, we test how well POD-Net can generalize to scenes with both novel objects and colors. We evaluate the generalization of our model on two datasets: a novel object dataset consisting of 20 new objects and a novel color dataset, where each object is split into two colors.

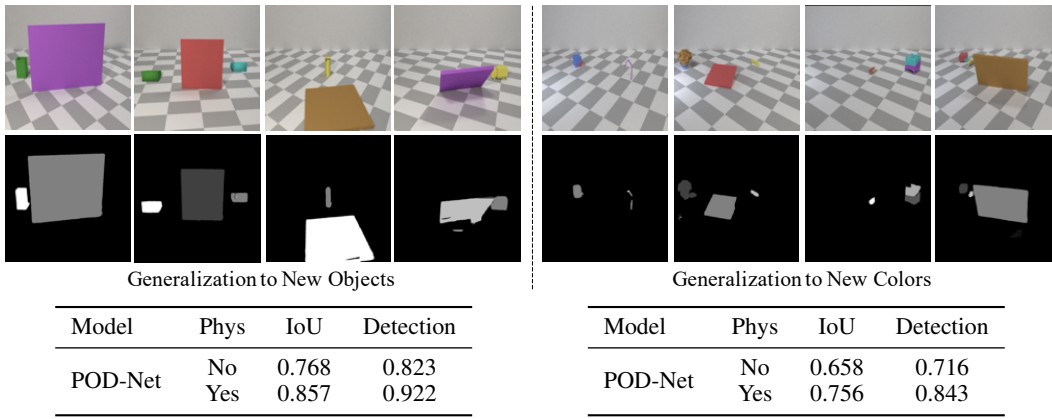

Generalization to New Objects

| Model | Phys | IoU | Detection |
|-------|------|-----|-----------|
| POD-Net | No | 0.768 | 0.823 |
| | Yes | 0.857 | 0.922 |

Generalization to New Colors

| Model | Phys | IoU | Detection |
|-------|------|-----|-----------|
| POD-Net | No | 0.658 | 0.716 |
| | Yes | 0.756 | 0.843 |

Figure 7: Generalization to novel objects and colors. Top: POD-Net successfully segments individual objects, except when colors bisect an object (row 2, column 7). Bottom: Evaluation of POD-Net's generalization with or without physical constancy, measured in average IoUs on segmentations and in the percentage of objects that are detected. Including physics integrates the motion signal and generalizes better in both cases.

Figure 7 shows a quantitative analysis of POD-Net applied to both novel objects and colors. We find that in both settings, POD-Net with physical consistency gets better segmentation than without. Performance is higher here compared to that reported on the training set, as both novel datasets contain fewer objects in a single scene. Qualitatively, POD-Net performs well when asked to discover novel objects, although it can mistake a multicolored novel shape to be two objects.

## 4.2 REAL BLOCK TOWERS

Next, we evaluate how POD-Net segments and detects objects in real videos.

**Data.** We use the dataset in Lerer et al. (2016) with 492 videos of real block towers, which may or may not be falling. Each frame contains 2 to 4 blocks of red, yellow, or green color. Each block has the same 3D shape, although the 2D projections on the camera differ.

**Setup.** For our backprojection model, we use a pretrained neural network on scenes of a single block at different heights, sizes, and varying distances. Similar to Section 4.1, the backprojection model serves as a rough 2D to 3D model and is trained with different camera and perspective parameters without occlusion. All other settings are the same as in Section 4.1.

**Results.** We compare masks discovered by POD-Net and baselines in Figure 8. We find that OP3 and MONet often misses blocks and also groups two blocks into a single object, leading to floating blocks in the air (Figure 8, 2nd row). Normalized cuts also suffers from a similar issue of grouping blocks, but suffers an additional issue of oversegmentation (see Appendix A.1). UVOD fails to predict a segmentation due to limited motion in video. POD-Net (single scale, no physics) is able to segment

| Model | Multi-Scale | Phys | IoU | Detection |
|-------|-------------|------|-----|-----------|
| MONET | - | - | 0.521 (0.005) | 0.537 (0.003) |
| OP3 | - | - | 0.311 (0.004) | 0.250 (0.007) |
| Norm. Cuts | - | - | 0.652 (0.006) | 0.849 (0.018) |
| UVOD | - | - | 0.029 (0.001) | 0.0 (0.0) |
| POD-Net | No | No | 0.546 (0.004) | 0.523 (0.006) |
| POD-Net | Yes | No | 0.734 (0.012) | 0.761 (0.008) |
| POD-Net | Yes | Yes | **0.837 (0.004)** | **0.908 (0.008)** |

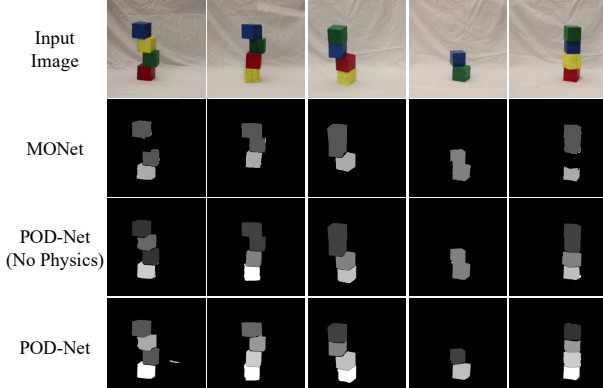

Figure 8: Top: IoU of segmentation results on the real blocks dataset and the percentage of objects detected. Bottom: Qualitative comparisons of unsupervised object segmentation of POD-Net with and without physics and with MONet on realistic block towers. MONet often groups two blocks of similar color (dark blue/green) together and sometimes misses particular blocks. POD-Net without physics reliably detects all blocks, but still groups similar blocks (dark blue/green) into one. POD-Net with physics detects all objects and assigns different masks to each. Standard error in parentheses.

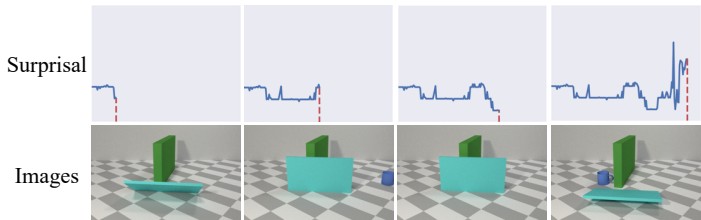

Figure 9: Surprise over time in a 'Block' scene. POD-Net has relatively low surprisal throughout most of the video. But when the occluder falls and the object appears to 'teleport' across the wall, POD-Net recognizes this abnormal shift in position and becomes surprised.

all blocks, but treats the entire stack as a single object. POD-Net (multi-scale, no physics) does better and is able to reliably segment all blocks, though it still groups blocks of similar colors together (Figure 8, 3rd row). Finally, POD-Net with multiple scales and physical consistency performs the best, reliably separating individual blocks in a tower (Figure 8, 4th row).

### 4.3 JUDGING PHYSICAL PLAUSIBILITY

We test whether POD-Net can discover objects reliably enough to perform the physical violation detection task of Smith et al. (2019), in which videos that have non-physical events (objects disappearing or teleporting) must be differentiated from plausible videos.

We consider two separate tasks: the Overturn (Long) task, which consists of a plane overlaying an object, and the Block task, which consists of physical scenes with a solid wall and an object moving towards the wall, where it may either appear to hit the wall and stop or appear on the other side. To successfully perform the Overturn task, POD-Net must reason about object permanence, while to accomplish the block task, the system must remember object states across a large number of timesteps and understand both spatial continuity and object permanence.

**Setup.** We use POD-Net trained in Section 4.1 to obtain a set of physical objects (represented as cuboids) describing an underlying scene. The extracted objects are provided as a scene description to the stochastic physics engine and particle filter described in Smith et al. (2019). We evaluate our models using a relative accuracy metric (Riochet et al., 2018).

**Results.** On the Block task, we find that our model achieves a relative accuracy of 0.622. Its performance on a single video can be seen in Figure 9, where it has learned to localize the block well enough that the model is surprised when it appears on the other side of the wall. The model in Smith et al. (2019) scores a relative accuracy of 0.680; this acts as an upper bound for the performance of our model, since supervised training is used to discover the object masks and recover object properties. In contrast, POD-Net discovers 3D objects in an unsupervised manner, outperforming the baseline generative models studied by Smith et al. (2019) that do not encode biases for objecthood (GAN: 0.44, Encoder-Decoder: 0.52, LSTM: 0.44). On the Overturn (Long) task – the one task where the ADEPT model underperforms baselines – our model obtains a performance of 0.77, outperforming Smith et al. (2019) (0.73), and equalling or exceeding models that do not encode biases for objects (GAN: 0.81, Encoder-Decoder: 0.61, LSTM: 0.63).

A limitation of our approach towards discovering 3D object primitives is that across a long video (over 100 timesteps), there may be several spurious extraneous objects discovered. The model in Smith et al. (2019) does not deal well with such spurious detections either, requiring us to tune separate hyper-parameters for each task.

## 5 CONCLUSION

We have proposed POD-Net, a model that discovers 3D physical objects from video using self-supervision. We show that by retaining principles of core knowledge in our architecture – that objects exist and move smoothly – and by factorizing object segmentation across sub-patches, we can learn to segment and discover objects in a generalizable fashion. We further show how these discovered objects can be utilized in downstream tasks to judge physical plausibility. We believe further exploration in this direction, such as integration of more flexible representation of physical dynamics (Mrowca et al., 2018; Sanchez-Gonzalez et al., 2020), is a promising approach towards more robust object discovery and a richer understanding of the physical world around us.

**Acknowledgements.** This work is in part supported by ONR MURI N00014-16-1-2007, the Center for Brain, Minds, and Machines (CBMM, funded by NSF STC award CCF-1231216), the Samsung Global Research Outreach (GRO) Program, Toyota Research Institute, and Autodesk. Yilun Du is supported in part by an NSF graduate research fellowship.

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

## A    APPENDIX

### A.1    NORMALIZED CUT/ CRISP BOUNDARY DETECTION QUALITATIVE VISUALIZATION

We provide visualizations of segmentations using normalized cuts in Figure A1. Normalized cuts relies on local color similarity and spatial locality to determine segments of objects. Since our objects area relatively similar in color, it is able to segment the rough shape of objects. However, compared to POD-Net, normalized cuts results in significantly less sharp boundaries around each shape and leads to over-segmentation of an individual object into multiple separate objects. This is due to the fact that shapes still exhibit variations in object color from lighting that causes normalizing cuts to incorrectly segment the object.

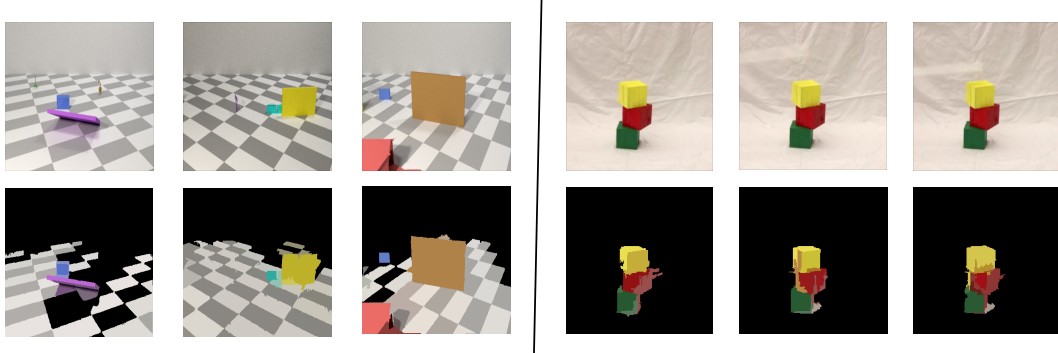

Figure A1: Illustration of segmentation using normalized cuts. Normalized cuts often over-segments objects and sometimes misses segmentation of small objects.

We further provide visualizations of segmentations using crisp boundary detection in Figure A2. Crisp boundary detection detects the set of edges that determines an objects. This enables the approach to segment large objects in a scene, where edges are not ambiguous, but fails to accurately segment smaller objects in a scene, which have more ambiguous edges. Furthermore, detected edges are also sensitive to the lighting of an object, sometimes segmenting an object into multiple separate pieces.

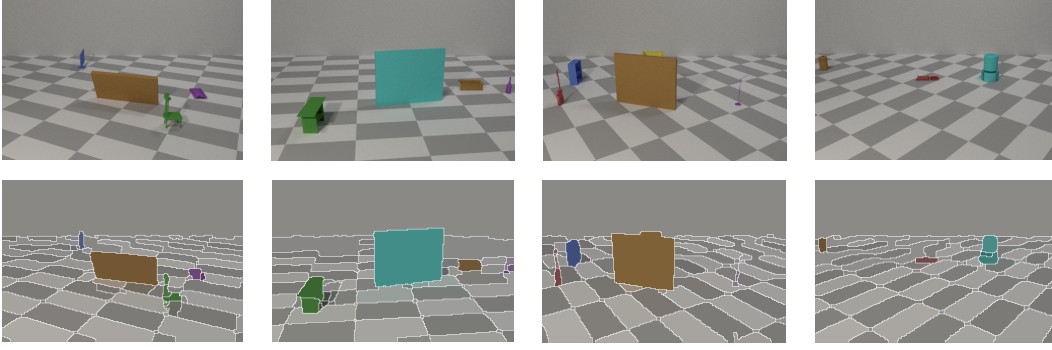

Figure A2: Illustration of segmentation using crisp boundary detection. Crisp boundary detection is able to segment large objects in a scene, but fails to accurately segment smaller objects in a scene, and sometimes segments objects to multiple small pieces.

### A.2    DETAILS ON MANUALLY DESIGNED PROJECTION MODULES

To manually design a backprojection and projection model, we assume each physical object $(\boldsymbol{t}_k, \boldsymbol{s}_k, \boldsymbol{q}_k)$ has a fixed rotation $q_k$ and z-axis length $s_{zk}$, and regress other size and position parameters. For our backprojection model, given a segmentation mask $\boldsymbol{m}_k$ of our object, we determine the 2D bounds of the mask $x_{\min}, x_{\max}, y_{\min}, y_{\max}$ in a differentiable manner, by taking the segmentation mask weighted mean of of boundary pixels (as defined by the 200 most boundary pixels in each direction). We then set $\boldsymbol{t}_z = 1 + \alpha y_{\min}$, corresponding to the assumption that higher segmentation masks correspond to further away objects. We compute the remaining coordinates of $\boldsymbol{t}_k$ and $\boldsymbol{s}_k$ by

inverting a camera intrinsic matrix on the 2D bounds of the mask and setting the center of an object to be halfway between the extremes the resultant coordinates. We note that our backprojection model is fully differentiable. For our projection model, we utilize the camera matrix to explicitly project inferred primitives back to the segmentation mask.

### A.3 PER CATEGORY SEGMENTATION PERFORMANCE

We report per category segmentation performance on ShapeNet objects below: pillow: 0.554, mug: 0.450, rocket: 0.347, earphone: 0.643, computer keyboard: 0.696, bus: 0.694, camera: 0.725, bowl: 0.565, bookshelf: 0.648, stove: 0.587, birdhouse: 0.544, wine bottle: 0.695, bench: 0.404, microwave: 0.314, lamp: 0.300, pistol: 0.566, chair: 0.465, cabinet: 0.646, bag: 0.602, rifle: 0.497, file: 0.467, faucet: 0.367, car: 0.594, bathtub: 0.500, microphone: 0.530, ashcan: 0.729, basket: 0.752, knife: 0.676, mailbox: 0.578, table: 0.565, printer: 0.660, cap: 0.559, sofa: 0.404, vessel: 0.404, display: 0.771, loudspeaker: 0.646, bicycle: 0.598, remote: 0.720, helmet: 0.368, train: 0.567, telephone: 0.601, jar: 0.663, piano: 0.734, washer: 0.358.

We find that despite having an internal physics representation of a cube, there is relatively little correlation between well-segmented objects and cubeness, with POD-Net performing well on non-cuboid classes such as piano while performing poorly on cuboid classes such as washer. Our physics loss instead encourages segmented objects across time to translate uniformly as well as maintain size.

### A.4 MODEL ARCHITECTURE

We detail our attention model in Table A1a and our component VAE model in Table A1b. In contrast to Burgess et al. (2019), we use a residual architecture for both attention and component VAE networks, with up-sampling of the spatial broadcast layer.

| 7x7 Conv2D, 32 |
| --- |
| BatchNorm |
| 3x3 Max Pool (Stride 2) |
| ResBlock Down 32 |
| ResBlock Down 64 |
| ResBlock Down 128 |
| ResBlock Up 256 |
| ResBlock Up 128 |
| ResBlock Up 64 |
| ResBlock Up 32 |
| ResBlock Up 32 |
| 3x3 Conv2D, Output Channels |

(a) Attention Model ($\alpha_\psi$)

| 7x7 Conv2D, 32 |
| --- |
| BatchNorm |
| 3x3 Max Pool (Stride 2) |
| ResBlock Down 16 |
| ResBlock Down 32 |
| ResBlock Down 64 |
| Global Average Pool |
| Dense $\rightarrow$ 256 |
| $256 \rightarrow 32$ ($\mu, \sigma$) |
| z $\leftarrow \mathcal{N}(\mu, \sigma)$ |
| Spatial Broadcast $z$ (8x) |
| 3x3 Conv2d, 256 |
| ResBlock up 128 |
| ResBlock up 64 |
| ResBlock up 32 |
| ResBlock up 16 |
| ResBlock up 16 |
| 3x3 Conv2D, Output Channels |

(b) VAE Component Model. ($q_\phi, p_\theta$)

Table A1: Overall Model Architectures used in POD-Net

We detail the architecture of our Backprojection and Projection Models in Table A2.

| Dense → 512 |
| --- |
| 512 → 1024 |
| View 64 x 4 x 4 |
| ResBlock Up 64 |
| ResBlock Up 64 |
| ResBlock Up 64 |
| ResBlock Up 64 |
| ResBlock Up 32 |
| 3x3 Conv2d, Output Channels |

| ResNet 18 |
| --- |
| Dense → 7 |

(a) Architecture of Backprojection Model

(b) Architecture of Projection Model

Table A2: Overall Model Architectures used in POD-Net

## A.5 COMPARISON ON PARTIALLY OCCLUDED OBJECTS

We further explicitly compare the performance of POD-Net on segmenting objects that occlude each other. We evaluate on the ADEPT dataset, but only consider objects such that the bounding boxes intersect. We find that in this dataset of objects, POD-Net (multi-scale, physics) obtains has a detection rate of 0.734, with the an average IoU of 0.701 while POD-Net (multi-scale, no physics) obtains a detection rate of 0.601 (IoU threshold 0.5) with an average IoU of 0.576. This indicates our approach in incorporating physics is able to learn to effectively separate objects that partially occlude each other.

