# OpenReview forum: "Unsupervised Discovery of 3D Physical Objects from Video"
_ICLR.cc/2021/Conference — ICLR 2021 Poster_

### Official Review · AnonReviewer3 · 2020-10-28
**Un-compelling experimental evaluation, missing details**

**Rating:** 6
**Confidence:** 4

**Review:**

This paper proposes an unsupervised approach for discovering 3D objects from video. Given training videos with objects moving around in a scene, the paper learns to decompose the given images into segments. Each segment is associated with a 3D location (translation, rotation and scale) as well as 3D dynamics (linear and angular velocity), and latent vector capturing the appearance. This appearance is rendered to generate images. A reconstruction loss on the generated images, along with a physics loss on the inferred object locations is used to train the model. The paper conducts experiments on videos rendered using the ShapeNet dataset, and videos of real block tower falling (from Lerer et al.). It measures the accuracy of predicted segments, and compares against other object discovery methods, as well as image segmentation methods. Lastly, the paper reports an experiment for using the proposed model to predict physical plausibility of video stimuli.

Strengths: The proposed unsupervised approach of learning physical properties of 3D objects from videos is novel to the best of my knowledge.

Shortcomings: My major issue is with the experimental evaluation. The primary metric used in the paper measures the intersection over union of predicted masks with ground truth masks. This doesn't evaluate the other properties that the paper claims to infer (3D geometry and position of each object). Thus, the current experimental evaluation falls short in evaluating all parts of the proposed model.

Furthermore, even if we limit to evaluation of segmentation masks, I am not sure if the paper is using appropriate baselines. There are a number of papers that tackle unsupervised edge detection (and consequently detection of object segments), see Isola et al. ECCV 2014, Unsupervised Learning of Edges, Li et al. CVPR 2016. A comparison to such unsupervised segmentation techniques should be made.

Lastly, the paper is missing details and hard to read: a) at test time, is the segmentation done on a per-frame basis or on the basis of the whole image sequence, b) what does the physics loss capture -- is it trying to induce a first order motion (zero acceleration) on the object, c) why is the learned physics model not used to judge the physical plausibility, d) the paper assumes a parametric form for objects (3D cuboids), but baselines it compares to likely don't (eg: normalized cut doesn't), why are experiments in Figure 7 a fair comparison?

In summary, I quite like the design of the unsupervised technique to learn about objects and their physical properties, I do not find the experiments convincing.

Update: I thank the authors for providing additional 3D metrics, and comparisons to unsupervised 2D segmentation techniques. I have updated my rating.

---

> ### Author Response · Authors · 2020-11-24
> **3D Evaluation and Comparisons**
>
> Thanks for the feedback. We have added 3D evaluation, and have also compared with recent unsupervised edge detection methods. We clarify the details below.
>
> **Q1) Evaluation in 3D**
>
> We agree and, as mentioned in the general response, have added evaluations of 3D object recovery in Table 1, where we measure 3D IOU and detection of recovered 3D objects. Since there is ambiguity in the recovered depth/scale of recovered objects, we first fit a linear regression between predicted depth/scale to ground truth depth and scale to determine a mapping from our inferred 3D coordinates and ground truth 3D coordinates. As mentioned in our general response, we have also included Figure 6, showing the correlations of inferred 3D location and size with respect to ground truth annotations of 3D location and size.
>
> **Q2) Comparison to unsupervised edge detection methods**
>
> Following your suggestion, we have added comparisons to unsupervised edge detection based on [1] in Table 1. We have also presented qualitative samples in A.1.1. We find that such an approach can reliably segment large images in our scene, where edges are not ambiguous, but fails for smaller objects where edges are more ambiguous (3rd image, yellow object in Figure A2). Furthermore, sometimes objects are segmented into multiple parts (2nd and 4th image in Figure A2), due to nonexistent edges that are made apparent by lighting.
>
> **Q3) At test time, is the segmentation done on a per-frame basis or on the basis of the whole image sequence?**
>
> At test time, segmentation is done on a per-frame basis. We have clarified this in Section 4.1
>
> **Q4) What does the physics loss capture -- is it trying to induce a first-order motion (zero acceleration) on the object?**
>
> Your understanding is correct, our loss is indeed trying to induce first-order motion, as well as shape constancy. We have clarified this in Section 3.2 (dynamics model)
>
> **Q5) Why is the learned physics model not used to judge the physical plausibility?**
>
> The goal of our learned physics model is only to discover a set of 3D objects that are consistent with the aforementioned qualities of first-order motion and shape constancy. By discovering 3D objects with such properties, we can then utilize a more complex physics engine to reason about the physical plausibility of an observed video over the 3D objects.
>
> **Q6) Results in Figure 7: the paper assumes a parametric form for objects (3D cuboids), but baselines it compares to likely don't (eg: normalized cut doesn't).**
>
> In Figure 7, it is true that POD-Net with physics assumes that the parametric form of a 3D object is a cube. However, we do not believe this is the main source of gains on our approach, since the loss primarily just enforces that shapes are consistent in size and translation across time. We illustrate this by reporting per category segmentation performance on the dataset evaluated in Section 3.1 in section A.1.3. We find that those well-performing classes are relatively uncorrelated with their cubeness, a characteristic we believe also carries over to Figure 7.
>
> [1] Phillip Isola, Daniel Zoran, Dilip Krishnan, and Edward H. Adelson. Crisp Boundary Detection Using Pointwise Mutual Information

---

### Official Review · AnonReviewer2 · 2020-10-28
**Interesting contribution, but confusing equations/text**

**Rating:** 6
**Confidence:** 4

**Review:**

Summary:
This paper proposes a deep network architecture to attack the important problem of discovering physical objects from observing videos only. The architecture is mainly an extension of the MONet architecture with a physics loss and a multi-scale inference scheme. The physics loss encourages the inference module (that masks out the objects) to be more consistent with the observed motion in the scene. Additionally, the multi-scale scheme, motivated by human foveation patterns, shows better empirical performance. The method outperforms reasonable baselines on both a synthetic dataset and a real-world dataset.

The immediately related works are summarized nicely. However, I believe it might be worth mentioning other methods that discover objects by using motion, such as motion segmentation. Traditional examples include [1], and more recent examples include [2,3,4].

Overall, I think the paper poses an interesting contribution with interesting results. However, the paper is hindered by the confusing equations and text, making it incredibly difficult to reproduce from the paper alone, and what I believe to be false claims of unsupervised learning (see weaknesses for details).

Strengths:
- Empirical results show that the method significantly outperforms reasonable baselines.
- The proposed physics loss is well-motivated and leads to improved empirical performance on both datasets. Additionally, the multi-scale scheme, while simple, surprisingly leads to large gains.
- The connection to violation-of-expectation is interesting, and the method shows pretty decent performance compared to the ADEPT upper bound.

Weaknesses:
- The equations and the text describing them are very confusingly written, leading to almost irreproducibility of the work from the text alone. See the questions section for my detailed questions on the math.
- The method claims that the method is unsupervised. However, the unprojection and projection models are pre-trained with supervision, and those models are required in order to evaluate $L_{physics}$. The text claims that the unprojection can be done assuming camera parameters and the plane height, but I don't see how this is the case. Additionally, the projection method, which computes an object mask only from translation/rotation/scale, cannot possibly be unsupervised as the input information lacks shape information. Thus, the claim seems to be false.

Questions:
- Questions about the equations:
    - Please define $alpha_{\psi}$. I assume it's the same as Attention(). Please keep the notation consistent.
    - In the dynamics model section, how is "foreground" defined for the indicator function? This indicator is supposed to be part of the function that predicts $\hat{m}^t_k$, which is the predicted foreground mask of object k. Please clarify this.
    - "closer than all other objects at the specified pixel location" <- what does this mean? Can multiple objects be present at the same pixel location? This doesn't appear to be the case when looking at the definition of $\mathbf{c}_i$ (context of the RNN).
    - Eq (4):
        - I believe the LHS is missing notation: $p(\mathbf{x}_i | \mathbf{z})$, as this equation should be for a single pixel. Or is it instead missing a product over i?
        - Where is $k$ in this sum?
        - The subscript on $m$ should be $k$, shouldn't it? Why is it $i$? Please clarify this.
        - $\sigma$ is being overloaded (it was used in the dynamics model section). Please introduce a subscript for clarity.
        - $c$ is also overloaded; this was context vector for the inference RNN.
        - Why is the background probability being multiplied, not added? It doesn't seem to be a valid probability otherwise.
        - Why does probability of $c$ depend on $m$? Shouldn't it depend on $z$? Additionally, it was never defined.
    - Loss function:
        - In $L_{image}$, what is Decode($\mathbf{x}^t$ | $\mathbf{z}_k^t$)? Is it equal to Eq. (4)? Please clarify this.
        - In $L_{image}$, what is Decode($\mathbf{m}^t_k$ | $\mathbf{z}^t_k$)? Is this meant to be Decode($\mathbf{c}^t_k$ | $\mathbf{z}^t_k$)? I'm also assuming $\mathbf{m}^t_k$ should be multiplied to that term (if so, please fix the parentheses for clarity).
        - The MONet loss uses a KL term to minimize the loss between the masks produced by the inference module ({$m_k$}) and the masks produced by the VAE decoder ({$c_k$}). According to my deductions of what $L_{image}$ is supposed to be, this now occurs there. Is there a reason for this choice?
- "After qualitatively observing object like masks..." is there a automatic way to do this? Having to baby-sit the training procedure is not ideal. Additionally, how many iterations is this?
- Is there any reason that POD-Net w/out multi-scale + physics outperforms MONet? The two architectures should be quite similar.
- What happens if there are inconsistent errors in the segmentation of the subpatches? Does the ordering of the subpatch processing matter?

Comments:
- The physics representation is quite rudimentary, which I imagine will limit the method to simple scenes such as the proposed Moving ShapeNet dataset. More complex scenes will require a higher-fidelity physics representation.
- If the math/text and claims of no supervision are fixed and/or addressed, I would be willing to increase my score.


[1] T. Brox and J. Malik. Object segmentation by long term analysis of point trajectories. In European Conference on Computer Vision (ECCV), 2010.

[2] P. Bideau, A. RoyChowdhury, R. R. Menon, and E. LearnedMiller. The best of both worlds: Combining cnns and geometric constraints for hierarchical motion segmentation. In IEEE Conference on Computer Vision and Pattern Recognition (CVPR), 2018.

[3] C. Xie, Y. Xiang, Z. Harchaoui, and D. Fox. Object discovery in videos as foreground motion clustering. In Proceedings of the IEEE Conference on Computer Vision and Pattern Recognition (CVPR), 2019.

[4] A. Dave, P. Tokmakov, and D. Ramanan. Towards segmenting anything that moves. In Proceedings of the IEEE International Conference on Computer Vision Workshops (ICCVW), 2019.

Update: Thanks for the authors' response. The authors have clarified the equations, addressed the concerns of claims of no supervision, and provided more experiments and metrics to back the current set of claims. I do believe this paper to be interesting enough for an ICLR paper, and have updated my score accordingly.
Also, do note that the motion segmentation works do not assume a single foreground object, they assume an arbitrary amount of them. However, you are correct in that they assume an entire video as input, as opposed to the proposed method which can segment individual frames.

---

> ### Author Response · Authors · 2020-11-24
> **Equations / Unsupervised Results**
>
> Thanks for the constructive comments. We have clarified the equations in the method section and have also revised our statements about the unsupervised nature of our approach.
>
> **Q1) Questions about the equations**
>
> We have rewritten the overall text and equations in the method section following your suggestion, and have renamed variables to prevent overloading. We have also updated our related work with your suggested references. We have revised our original formulation of $L_{\text{image}}$ and have moved the KL difference between masks to $L_{\text{kl}}$, while using $L_{\text{image}}$  to directly maximize the likelihood in Equation 4. We have factorized Equation 4 to be across individual pixels in an image, and have also defined the missing equations.
>
> **Q2) Questions about weak supervision**
>
> You are right that we cannot infer rotation, shape, and translation information all directly from a hand-coded unprojection model with camera matrices, and we have revised the text to say this. However, we are able to infer the rough $x, y, z$ position of an object, and its $x, y$ size, by inferring depth ($z$) from the bottom coordinate of an object and inferring remaining values by inverting the camera matrix. We construct such an unprojection model, which we report in Table 1 (and detail our implementation in Appendix A.1.2), and set the rotation and $z$ size of objects to be constant. We find that such an unprojection model still leads to good segmentation performance. We note that unsupervised approaches also assume domain knowledge when doing unsupervised 3D segmentation, such as using plane normals matching and convex/concavity assumptions for object shape [1]. Nevertheless, we now explicitly note that our approach uses supervised information in terms of 2D to 3D mapping in the last paragraph of the related works.
>
> **Q3) Related work on motion segmentation**
>
> Thanks for your pointers to related works on motion segmentation. We have added them to our related work. Note that at test time, these models require a video as input, while our model requires only a single image. This is because they study a fundamentally different problem---segmenting a single foreground moving object into several component moving directions---while we focus on learning segment objects using motion cues.
>
> **Q4) "After qualitatively observing object-like masks..." is there an automatic way to do this? How many iterations?**
>
> We have clarified that this is approximately after 100,000 training iterations in Section 3.3. Instead of qualitatively observing object-like masks, we can instead train models until convergence.
>
> **Q5) Why POD-Net w/out multi-scale + physics outperforms MONet?**
>
> POD-Net without either multi-scale cues or physics slightly outperforms MONet due to the underlying architecture. MONet uses a spatial broadcast of initial latents to the full desired output resolution, while POD-Net uses a series of residual blocks to upsample images. This residual architecture allows for finer segmentations.
>
> **Q6) Does the ordering of the subpatch processing matter?**
>
> In settings in which there are inconsistencies in different segmentations in overlapping subpatch regions, we take the OR of either segmentation, and merge different segments together if such a result causes two distinct segments to share masked pixels. Due to this overlapping procedure, our approach is invariant to the order of subpatches.
>
> [1] Karpathy et al. Object Discovery in 3D scenes via Shape Analysis. ICRA 2013

---

### Official Review · AnonReviewer1 · 2020-10-29

**Rating:** 6
**Confidence:** 3

**Review:**

The paper introduced an unsupervised approach for 3D physical object segmentation from the video. The proposed algorithm decomposes the scene into several 3D primitive shapes, and learn to generate latent descriptors for a generated model to reason the 3D properties. The system also enforces assumptions that object move according to physics, and eventually produce a scene representation in a self-supervised manner.

The paper is well motivated and the proposed approach and design are both novel. Experimental results suggest that the proposed approach is effective and the ablation study is helpful too. Moreover, the writings of this paper are clear and the visualizations and diagrams in the paper are informative and easy to follow.

I have some additional comments regarding the paper:
- In table 1,  the normalized cuts method is a very classical algorithm, but the performance is very good. I was wondering whether authors can provide a few more explanations.
- Moreover in table 1, why “percentage of objects detected in an image” is so low? (below 1%) I wonder if I missed anything from the descriptions of the paper.
- In the paper, the approach is under the assumption that each object of the scene is rigid. I understand it's beyond the scope of the paper but the world also consists of many elastic objects and infants are likely to discover those objects well. I wonder if there is a way to extend the system to handle such senarios.
- Finally, at a high-level, the authors discussed in the related work section that the proposed work is different from existing self-supervised segmentation methods. I think the claims in those section makes sense, but I was wondering if authors can provide some comments on the performance of POD-Net on more challenging scenes (maybe for more complicated backgrounds as a starting point) or if there is a way to compare with existing self-supervised segmentation methods.

---

> ### Author Response · Authors · 2020-11-24
> **Clarifications/Comparisons**
>
> Thank you for your detailed comments. We have added clarifications about normalized cuts and about underlying assumptions and results. We have also added an additional self-supervised segmentation method and have also revised the submission accordingly.
>
> **Q1) Explanations on the results of normalized cuts**
>
> Normalizing cuts rely on a trade-off between color segmentation and spatial locality to segment objects. As a result, normalizing cuts performs relatively well in the settings we consider, since many of our objects are differentiated from each other primarily by color.  Since objects in our scenes are relatively monochrome, this enables normalized cuts to segment most of the identity of an object from an image. However, due to shadow and lighting effects, normalized cuts are not able to capture boundaries of objects and also sometimes over-segment objects, since different locations of objects still have relatively different colors. This can also be seen in Figure 4. We have added a more detailed  description of normalized cuts in Appendix A.1.1.
>
> **Q2) In Table 1, why “percentage of objects detected in an image” is so low? (below 1%)**
>
> Our apologies, we mistakenly added a percentage indicator in our table. We actually report the proportion of objects detected in images (with object detection corresponding to performance of 1.0). We have removed the percentage indicator in our tables.
>
> **Q3) In the paper, the approach is under the assumption that each object of the scene is rigid. I understand it's beyond the scope of the paper but the world also consists of many elastic objects and infants are likely to discover those objects well. I wonder if there is a way to extend the system to handle such scenarios.**
>
> We agree this would be an interesting future direction to explore. We see our approach as proposing a general paradigm to utilize physics to help discover 3D objects. To utilize our framework for elastic objects, we can replace objects with graph particles and use particle dynamics predicted by a graph neural network [1, 2]. We have added this to the discussion section.
>
> **Q4) Finally, at a high-level, the authors discussed in the related work section that the proposed work is different from existing self-supervised segmentation methods. I think the claims in those sections make sense, but I was wondering if authors can provide some comments on the performance of POD-Net on more challenging scenes (maybe for more complicated backgrounds as a starting point) or if there is a way to compare with existing self-supervised segmentation methods.**
>
> Thanks for the suggestion. POD-Net's underlying 2D segmentation algorithm is based on MONET. As such probabilistic frameworks struggle with complex texture details, it has impacted POD-Net in more challenging scenes. We'd like to add that, on our datasets, we have compared with UVOD (an existing self-supervised segmentation method) and found that our approach generally does better. We also have added comparisons to Crisp Edge Detection, a separate class of self-supervised segmentation methods based on edge detections, and also found that our approach does better.
>
> [1] Damian Mrowca, Chengxu Zhuang, Elias Wang, Nick Haber, Li Fei-Fei, Joshua B. Tenenbaum, Daniel L. K. Yamins Flexible Neural Representation for Physics Prediction
>
> [2] Alvaro Sanchez-Gonzalez, Jonathan Godwin, Tobias Pfaff, Rex Ying, Jure Leskovec, Peter W. Battaglia Learning to Simulate Complex Physics with Graph Networks

---

### Official Review · AnonReviewer4 · 2020-10-30
**A so-called 3D object discovery method without any 3D evaluation**

**Rating:** 5
**Confidence:** 4

**Review:**

The paper proposed an unsupervised learning model, POD-Net, that learns to discover objects from video. The authors develop an inference model that performs image segmentation and object-based scene decompositions on overlapping sub-patches, and a generative model, which contains an unprojection step, a constant velocity dynamic model and a VAE, to reconstruct the original scene. With the novel dynamic model to predict motions for the 3D object-primitives, the POD-Net learns to segment objects with better physics.

++ Strong points:
The paper explores the use of motion cues to train a self-supervised model to extract object-based scene representations from videos. And in the approach that to unproject 2d masks into 3d primitives to compute the motion is novel to me, it allows the proposed approach to better discover objects with physical occupancy on the 2D videos.

Overall, the paper is well written. In particular, the intuition behind the method is described well. The Method section, the POD-Net, is easy to read and understand. And the Evaluation section is well structured, it is clear how the models are setted on different datasets.

++ Concerns:

The main concern on the paper is that although it claimed itself as a 3D object discovery method, all its evaluations are done on 2D datasets with 2D metrics. Although there are some reasonable improvements shown on these metrics, we do not know what is the capability of this work in terms of recovering 3D segmentation mask and 3D pose. This I consider incomplete for a work that claims its main difference w.r.t. prior work to be getting to 3D object discovery.

There have been a significant amount of prior work on unsupervised 3D object discovery (many of them on RGB-D) that is missed by the authors:

Herbst et al. Toward Object Discovery and Modeling via 3-D Scene Comparison. ICRA 2011
Karpathy et al. Object Discovery in 3D scenes via Shape Analysis. ICRA 2013
Ma and Sibley. Unsupervised Dense Object Discovery, Detection, Tracking and Reconstruction. ECCV 2014

Datasets such as

Lai et al. A Large-Scale Hierarchical Multi-View RGB-D Object Dataset. ICR 2011
Georgakis et al. Multiview RGB-D Dataset for Object Instance Detection. 3DV 2016

exist and they provide ways to evaluate unsupervised 2D-3D object discovery (one can start from RGB and deduct 3D pose and velocity). So I don't think the authors have enough excuses to not show any 3D results.

The author claims that the POD-Net is an unsupervised method, meanwhile bashing other methods of using pre-training (last paragraph of Section 1). However, their unprojection model and the project model is pre-trained and it is the same kind of supervision as the pre-trained segmentation models in other work.

Minor concerns:

-- In Sec 3.3, "model surprisal" doesn't sound to me like proper English, or maybe I'm missing something. If you are introducing a new phrase as a term you probably want to define it first.

-- In Sec 3.1 and 3.2, it is not clear how the author counts object discovery performance w.r.t. the time dimension, e.g. how is it handled if the ground truth mask is completely occluded? Is tracking consistency taken into account?

---

> ### Author Response · Authors · 2020-11-24
> **3D Evaluation**
>
> Thank you for your detailed comments. We have changed our related work, added 3D evaluations, and have addressed reducing supervision.
>
> **Q1) Prior work on unsupervised 3D object discovery**
>
> Thanks for referring us to related work, which we have cited and discussed in the revision. As you mentioned, the biggest difference between our work and these papers is that ours is based on RGB images. In contrast, [1] uses RGBD images and robotic manipulation, [2] also assumes a 3D mesh, while [3] builds up a 3D from RGBD images. The presence of depth significantly changes the overall difficulty of the unsupervised 3D object discovery problem; by knowing the underlying 3D structure, in many scenes, 3D objects can be easily segmented out in an unsupervised manner by noticing disparity in depth or edge/normal orientations and plane normals, especially since each paper considers a tabletop setting.
>
> **Q2) Datasets that provide ways to evaluate unsupervised 2D-3D object discovery (one can start from RGB and deduct 3D pose and velocity)**
>
> We agree. While both datasets focus on 3D object discovery from RGB-D input, instead of RGB input, their metrics for evaluating 3D discovery are useful. Following your suggestion, we have added evaluations of 3D object recovery in Table 1, using metrics in [4], where we measure the average 3D IOU between objects and the recall rate of recovered 3D objects. Since there is ambiguity in the recovered depth/scale of recovered objects, we first fit a linear regression between predicted depth/scale to ground truth depth and scale to determine a mapping from our inferred 3D coordinates to ground truth 3D coordinates. As mentioned in our general response, we have also included Figure 6, showing the correlations of inferred 3D location and size with respect to ground truth annotations of 3D location and size.
>
> **Q3) Supervision contained in the learned 2D to 3D mapping**
>
> In Table 1, we show that our learned projection model can instead be replaced with a hand-coded 2D to 3D geometry decoder, with small losses in performance. We detail our hand-coded 2D to 3D geometry decoder in Appendix A.3.  While our learned projection models serve as a type of domain knowledge, they do not assume any information about the underlying detection of an object. We note that other unsupervised approaches also assume domain knowledge when doing unsupervised 3D segmentation, such as matching plane normals and convex/concavity assumptions for object shape [2].  Nevertheless, we have added an explicit statement at the end of Section 1 discussing our use of supervised information, but note our approach can generalize to new shapes and sizes of objects.
>
> **Q4) Minor Concerns**
>
> In Sections 3.1 and 3.2, we report performance on all visible objects at the current timestep of the image, as both our approach and baselines only take a single image as input. “Model surprisal” the measured surprise of the model at each timestep. We have removed the mention of “model surprisal” in the text, and replaced it with “Surprise over time”.
>
> [1] Ma and Sibley.  Unsupervised Dense Object Discovery, Detection, Tracking and Reconstruction. ECCV 2014
> [2] Karpathy et al. Object Discovery in 3D scenes via Shape Analysis. ICRA 2013
> [3] Herbst et al. Toward Object Discovery and Modeling via 3-D Scene Comparison. ICRA 2011
> [4] Georgakis et al. Multiview RGB-D Dataset for Object Instance Detection. 3DV 2016

---

### Author Response · Authors · 2020-11-24
**Reviewer Response**

We thank all reviewers for their thorough feedback. Reviewers noted that the paper was well written [R1, R4], and that the proposed approach was novel and interesting [R1, R2, R3, R4]. We first address three major concerns shared by several reviewers: 3D evaluation [R3, R4], comparisons with unsupervised segmentation methods [R1, R3], and questions about the unsupervised nature of the framework, in particular the use of a learned 2D to 3D mapping [R2, R4]. We then respond to specific questions in individual responses below.

In response to requests for 3D evaluation, we have added metrics for 3D geometry estimation based on 3D IoU overlap and 3D recall to all methods in Table 1, using our unprojection module to infer 3D shapes. In the original Appendix, we have included a figure (Figure 10), showing the correlations of inferred 3D location and size with respect to ground truth annotations of 3D location and size. We have moved the figure to the main paper as Figure 6.

In response to requests for comparisons with unsupervised segmentation methods, we have added a comparison to an unsupervised edge detection approach, Crisp Edge Detection in Table 1, with a discussion of results in Appendix A.1.1.

Regarding the use of  supervision for 2D to 3D mapping, we have revised the last paragraph of the “related work” section to explicitly say that we use supervised information in terms of a learned mapping. At the same time, we have shown in Table 1 that we can use a hand-crafted 2D to 3D mapping, and still obtain good performance; we describe this implementation in Appendix A.1.2.

To allow everyone to easily see changes in the text, we have highlighted revised text in blue.

---

### Decision · Program_Chairs · 2021-01-07
**Final Decision**

**Decision:**

Accept (Poster)

**Comment:**

The paper proposes to do unsupervised discovery of 3D physical objects. The core idea is to decompose the scene into primitives that contain: (a) a segment; (b) 3D position and dynamics; and (c) appearance. These are combined with a physics model and renderer to discover objects/primitives by watching videos; the core supervisory signal used is that one should be able to reconstruct future scenes and that objects/primitives ought to be physically consistent. The system is tested on synthetic data as well as real videos of blocks.

The reviewers were positive about many aspects but, at the time of submission had a number of concerns. These were, in view of of many of the four reviewers, largely addressed. These are as follows:
- One overarching concern (R3, R4) was the experiments that the paper’s title and motivation focused heavily on 3D but the experiments lacked a 3D experiment of any variety. The authors addressed this by adding 3D IOU and recall. While numbers are low for IOU, this is a challenging area and the AC appreciates this as did R3 and R4.
- Another concern is the data itself (R4,R1). R4 in particular cites the synthetic nature of it as a stumbling block; R1 is similarly concerned about the difficulty of the backgrounds (and the rigidity of the objects). The AC thinks that the data is sufficient for this paper given the overall paper focus, methodological contributions, and particular set of claims. However, the AC is highly sympathetic to R4’s arguments and thinks more realistic real data (beyond the additional data of towers of blocks in front of a white sheet) would substantially improve the impact of the paper and the direction of research.
- The last content-focused concern was disagreement that the system is unsupervised (R2,R4). The authors have addressed this with experiments using a hard-coded system that uses a heuristic based on the bottom coordinate, which obtains good results as well. All reviewers with this concern seem satisfied although the AC would note this assumes a single ground plane, which ties into concerns about the data (although this is a small nitpick).
- R2 had substantial concerns about the legibility and reproducibility of the paper. These have been largely addressed in the revision, as far as the AC can tell.

The paper is an good contribution on a challenging and important problem. While the AC shares some of R4’s concerns about the data (and indeed how data difficulty and method interact), the AC finds the revised paper compelling and recommends acceptance.